# A simple method to isolate fatty acids and fatty alcohols from wax esters in a wax-ester rich marine oil

**Pauke Carlijn Schots**⊙*, **Guro Kristine Edvinsen, Ragnar Ludvig Olsen**

Norwegian College of Fishery Science, Faculty of Biosciences Fisheries and Economics, UiT The Arctic University of Norway, Tromsø, Norway

* pauke.schots@uit.no

## Abstract

*Calanus finmarchicus* is one of the most important zooplankton species in the North Atlantic. The zooplankton is currently being harvested and industrially processed to a marine oil product for human consumption as a marine nutraceutical containing long-chain omega-3 poly-unsaturated fatty acids. This oil is very rich in wax esters, a lipid class where fatty acids are esterified to long chain fatty alcohols. In this paper we describe a simple method to 1) isolate the wax esters from the other lipid classes present in the oil, 2) hydrolyze the wax esters, and 3) separate the fatty acids from the fatty alcohol, all by means of solid phase extraction. Starting with an average of 322 mg Calanus oil, we obtained 75 mg fatty alcohols and 63 mg fatty acids. Contrary to previously described techniques, our method neither oxidize the fatty alcohols to fatty acids, nor are the fatty acids methylated, allowing the native, unesterified fatty acids and fatty alcohols to be used for further studies, such as in cell culture experiments to study the metabolic effects of these specific lipid fractions rather than the intact oil or wax esters.

**Data Availability Statement:** All relevant data are within the paper and its Supporting Information files.

**Funding:** The publication charges for this article have been funded by a grant from the publication

## Introduction

The limited amount of fish oil, containing health promoting long-chain omega-3 polyunsaturated fatty acids (LC n-3 PUFA), available for the use in fish feed and human consumption, has led to a massive amount of research and development to find alternative sources for these fatty acids [1, 2]. Suggested possibilities have been to utilize lower trophic level alternatives, such as zooplankton, like krill and calanoid species, or the industrial cultivation of marine microalgae [3]. The marine copepod *Calanus finmarchicus* is one of the most important zooplankton species, with respect to biomass, in the North Atlantic and plays a key role in the pelagic food web between primary producers and economically important fish species [4–6]. Due to *C. finmarchicus*' importance in the marine ecosystem and the possibility to harvest it in a sustainable manner [7], the species has received substantial scientific attention. The zooplankton is currently being harvested and industrially processed to an oil (Calanus oil). [8, 9]. This oil, as most other marine oils, is a source of the health promoting LC n-3 PUFA [8, 9].

fund of UiT The Arctic University of Norway. The funders had no role in study design, data collection and analysis, decision to publish, or preparation of the manuscript.

**Competing interests:** The authors have declared that no competing interests exist.

However, unlike other marine oils, most of the fatty acids in Calanus oil (as much as 85%) are esterified to a fatty alcohol, forming a lipid class known as wax esters [8, 9].

Several analytical methods have been developed to identify and quantify the composition of the fatty acids and fatty alcohols in calanoid wax esters. These methods often contain transesterification, derivatization, separation, and purification steps of the wax esters and fatty acids and alcohols and lead to the production of fatty acid methyl esters (FAMEs) [10–13].

Previous studies have shown that the oil from *C. finmarchicus*, and especially the wax esters, have potent effects on gut microbiota and cardiac function and against the development of obesity, insulin resistance and inflammation in mice [14–18] and body composition, functional strength, cardiorespiratory function, metabolic markers and omega-3 index in humans [19–22], making this oil a possible new marine drug against metabolic syndrome and other pathological conditions.

After consumption, wax esters are to a large extend hydrolyzed to free fatty acids and free fatty alcohols in the lower end of the gastrointestinal tract [23]. The possible health effects of the wax ester derived fatty acids and fatty alcohols have been reviewed previouslyand it is known that both fatty acids and fatty alcohols can have positive health effects but potentially through different mechanisms [24]. To study the health effects of the individual lipid classes in the wax ester, in vitro, it is necessary to separate the fatty acids and fatty alcohols in hydrolyzed wax esters. To our knowledge there is no method previously described to separate the fatty acids and fatty alcohols in wax esters without the formation of FAMEs. Although the production of FAMEs is excellent for lipid composition analyses, it is less suitable for the extraction of specific lipid classes. FAMEs are esters of their own and would not permit us to obtain the free fatty acids without another saponification step. Similarly, the fatty alcohols would be oxidized to their corresponding fatty acids before being transformed to FAMEs. Thus, the production of FAMEs would not allow us to obtain the fatty alcohols and fatty acids in their native form without additional steps. Therefor the aim of this work was to develop a simple semi-preparative method using solid phase extraction (SPE), to isolate the wax esters from the other lipid classes present in the oil, and subsequently to separate the native non-esterified fatty acids and fatty alcohols in the wax esters. The method does not require sophisticated instrumentation. These isolated compounds may then be used to study potential metabolic effects in, for example, cell culture experiments.

## Materials and methods

### Oil and chemicals

The oil produced from *C. finmarchicus* (Calanus oil) was provided by the company Zooca® formerly named Calanus® AS (Tromsø, Norway). Heptane (99,8%), isopropanol (100%), diethyl ether (≥99,8%) and ethanol (96%) were purchased from VWR, Darmstadt, Germany. Acetic acid (99,8%) and hydrochloric acid (37%) were purchased from Honeywell, Seelze, Germany. Chloroform (99,0–99,4%), phosphoric acid 99,99%, potassium bicarbonate ≥99,95%, sodium hydroxide, copper (II) sulphate pentahydrate ≥98,0%, sulphuric acid and toluene were obtained from Sigma-Aldrich, Darmstadt, Germany. N-Hexane was obtained from Merck KgaA, Darmstadt, Germany.

### Isolation of wax esters from Calanus oil

The wax esters were isolated from Calanus oil through solid phase extraction (SPE) as described by Vang *et al.* [9] with some modifications. On average 322 mg of Calanus oil were dissolved in 6 mL chloroform (Fig 1). A Mega Bond Elute (5g) aminopropyl SPE disposable column (Agilent Technologies, Oslo, Norway),used in all separation steps described in Fig 1,

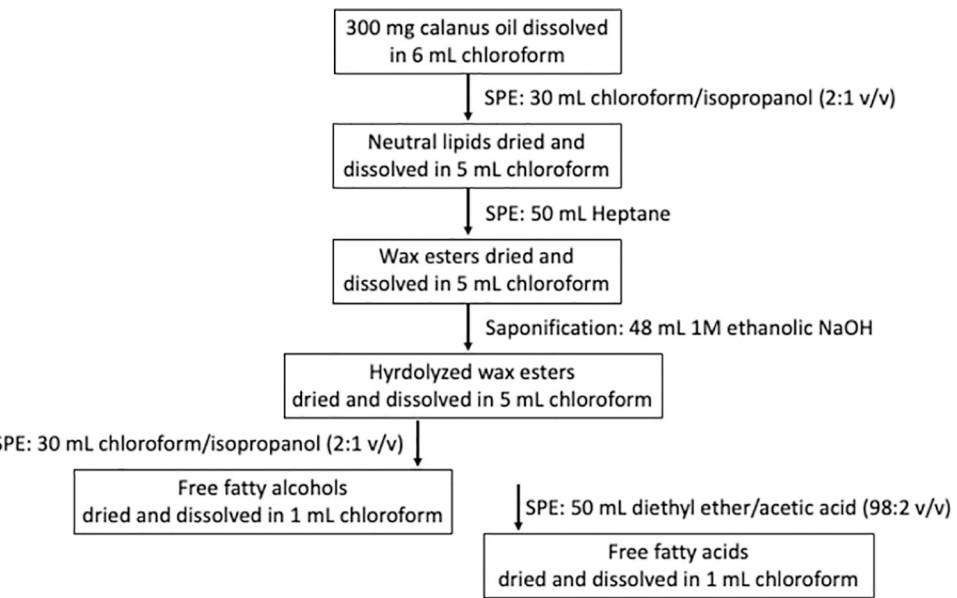

**Fig 1. Isolation of fatty acids and fatty alcohols in wax esters present in Calanus oil by use of solid phase extraction.** See text for details.

was mounted on a SPE Visiprep vacuum manifold (Supelco, Bellafonte, PA, USA). The column was conditioned with 20 mL heptane at a flow rate of approximately 1 mL/min, a flow rate used in all separation steps described in Fig 1. After conditioning, the Calanus oil in chloroform was added to the column. The neutral lipids (NL) present in the oil were eluted with 30 mL chloroform/isopropanol (2:1 v/v) and dried under nitrogen gas. The dried NL were subsequently dissolved in 5 mL chloroform and then applied to a new SPE column pre-conditioned with 20 mL heptane. The wax esters were eluted with 50 mL heptane and again evaporated to dryness under nitrogen gas and dissolved in 5 mL chloroform. The yields of the eluted lipid classes were determined gravimetrically after drying beforethey were redissolved in chloroform to provide a sample for the HPTLC analysis. The isolation of wax esters and the subsequent saponification and isolation of the native fatty acids and fatty alcohols were carried out 3 times (see S1 Table).

## Saponification of wax esters and isolation of the fatty acids and fatty alcohols

The isolated wax esters were saponified following Christie and Han [25] with slight modifications. The wax esters, on average 251 mg dissolved in 5 mL chloroform, were evaporated to dryness and dissolved in 50 mL of 1 M ethanolic NaOH to a concentration of 5 mg/mL. The dissolved wax esters were transferred to 5 GL 18 Duran Schott test tubes (DWK life sciences, Oslo Norway) each containing 10 mL dissolved wax esters in 1 M ethanolic NaOH, capped, and placed on a heating block (ThermoFisher Scientific, Oslo, Norway) for 90 min at 90˚C. No antioxidants were added during the wax ester hydrolysis because Calanus oil contains naturally the antioxidant astaxanthin. The hydrolyzed wax esters were cooled on ice and the Duran Schott test tubes were pooled before addition of 50 mL MilliQ water and 25 mL 6 M HCl. The acidified hydrolyzed wax esters were then mixed with 25 mL heptane to form two phases. The upper phase, containing the lipids, was collected and again evaporated under nitrogen gas and dissolved in 5 mL chloroform.

After the saponification, the free fatty acids and fatty alcohols were separated from each other by use of SPE (Fig 1). An SPE column was conditioned with 20 mL heptane after which the fatty acids and fatty alcohols in chloroform were added to the column. The fatty alcohols were eluted with 30 mL chloroform/isopropanol (2:1 v/v). The same column was then again conditioned with 20 mL heptane and the free fatty acids were eluted with 50 mL diethyl ether/acetic acid (98:2 v/v). The solvents with the separated fatty alcohols and fatty acids were evaporated under nitrogen gas and each dissolved in 1 mL chloroform. The use of SPE to separate the fatty alcohols from the fatty acids was chosen over the separation of the fractions via soap formation by NaOH, since the latter method did not lead to a clear separation in our set-up.

## Analysis by HPTLC

From the first isolation, aliquots of 50 μL were removed from the initial Calanus oil in chloroform and the fractions eluted from the SPE columns (Fig 1); the neutral lipids, the wax esters, the hydrolyzed wax esters and the isolated fatty alcohols and fatty acids, all in chloroform. The aliquots were stored at -20°C before analysis by high performance thin layer chromatography (HPTLC). One microliter of the samples was applied to the HPTLC plate (Silica gel 10 cm × 10 cm, Merck, Darmstadt, Germany) together with reference standard 18–5 A (Nu-Chek-Prep, Elysian, MN, USA). The plate was placed in a glass chamber saturated with heptane/diethyl ether/acetic acid (80:20:2 v/v/v) as described by Henderson and Tocher [26]. The mobile phase was allowed to migrate about 9 cm before the plate was removed and the solvent was allowed to evaporate. After air drying, the plate was sprayed with 10% copper sulphate in 8% phosphoric acid prior to development in an incubator at 180°C for 3 min. The developed HPTLC plate was scanned on a Xerox WorkCenter 7855i (Fig 2).

## Composition of the fatty acids and fatty alcohols

To analyze the composition of the wax ester, free fatty acids and fatty alcohols, aliquots of these lipid fractions from the first extraction round were dissolved in 1 mL toluene and 2 mL 1% sulfuric acid in methanol was added. The tubes were then flushed with nitrogen gas, capped, and incubated at 50°C for 16 h. After cooling, 2 mL 2% KHCO$_3$ was added, followed by 10 mL freshly made n-hexane/diethyl ether (1:1 v/v). After centrifuging at 1500 rpm for 2 min, the organic layer was transferred to a clean tube. The aqueous layer was mixed for a second time with 10 mL n-hexane/diethyl ether (1:1 v/v) and centrifuged before the organic layer was transferred to the organic phase from the first extraction. The combined organic phases were then dried by evaporation under nitrogen gas. The final extract was dissolved in n-hexane, to a final concentration of 1 mg lipid per ml. The fatty alcohols and the methylated fatty acids were determined by gas chromatography analysis on GC-FID (Agilent Technologies GC 7890B, Santa Clara, CA, USA) and identified by use of the FAME standards GLC 68D and GLC 96, and the fatty alcohol standard GLC 621 (Nu-Chek-Prep, Elysian, MN, USA). GLC621 is a combination of C14:0 myristyl alcohol (15% wt/wt), C16:0 palmityl alcohol (15% wt/wt), C16:1 palmitoleyl alcohol (15% wt/wt), C18:1 oleyl alcohol (15% wt/wt), C20:1 11-eicosenol alcohol (10% wt/wt), C21:0 methyl heneicosanoate (20% wt/wt), and C22:1 erucyl alcohol (10% wt/wt). This analysis was performed by the company Akvaplan-niva, daughter company of the Norwegian Institute for Water Research (NIVA), Tromsø, Norway (https://www.akvaplan.niva.no/en), following an inhouse protocol based on methods described by Folch et al., [27] and Christie [28].

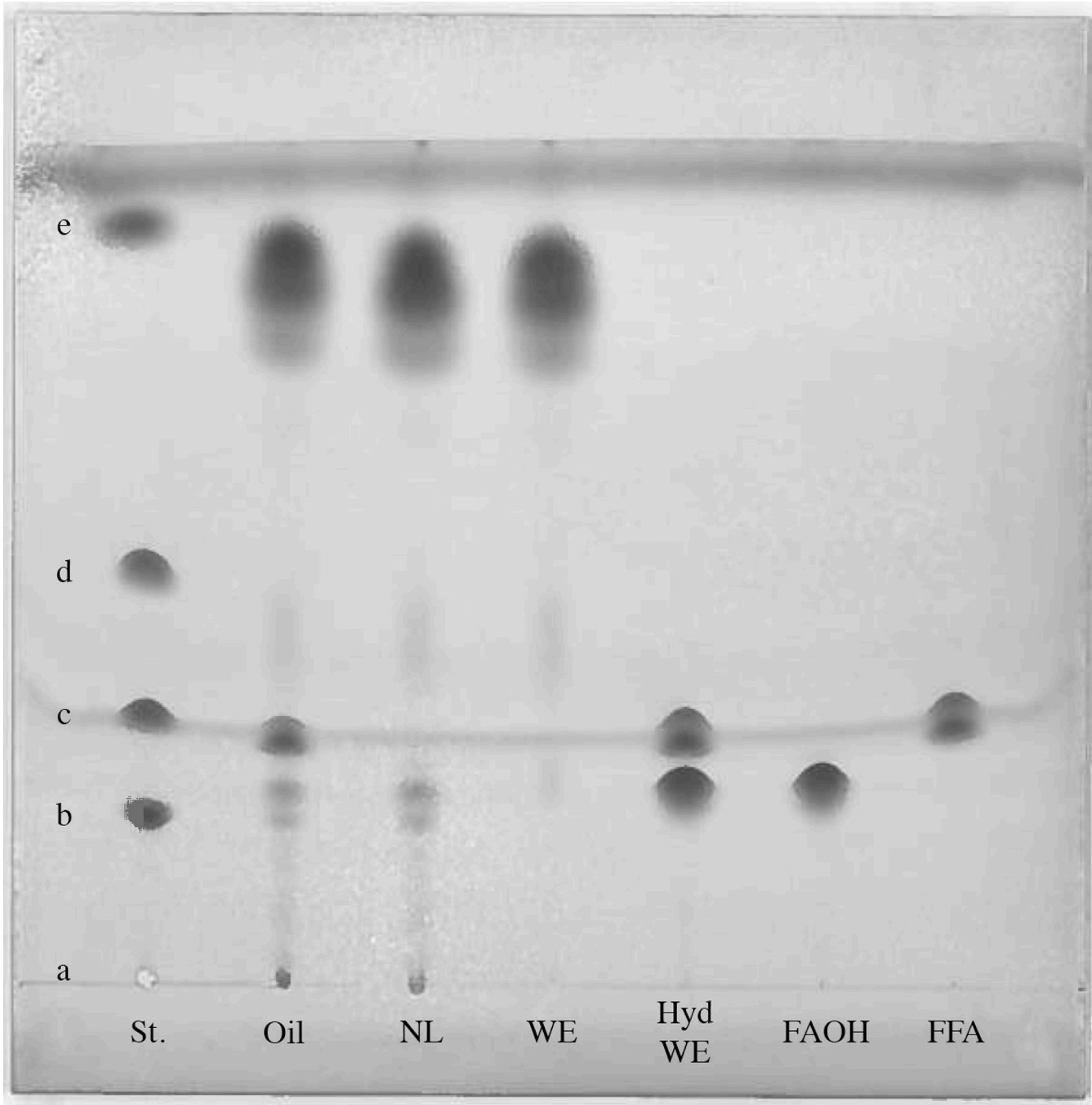

**Fig 2. High performance thin layer chromatography of the lipid classes isolated from Calanus oil by SPE.** St.: Lipid class standard 18–5 A containing lecithin (a); cholesterol (b); oleic acid (c); TAG (d); and cholesteryl oleate (e). Oil: Oil from *C. finmarchicus* (Calanus oil), NL: Neutral lipids, WE: Wax esters, Hyd. WE: Hydrolyzed wax esters, FAOH and FFA: free fatty alcohols and free fatty acids, respectively, from the hydrolyzed wax esters.

## Results and discussion

### Wax ester isolation

A flow diagram of the different steps to isolate the fatty acids and fatty alcohols of the wax esters in the Calanus oil by means of solid phase extraction (SPE) is shown in Fig 1. The total weight of wax esters (WE) isolated from on average 322 mg Calanus oil was 251 mg (Table 1 and S1 Table), i.e. constituting almost 80% of the total lipids in the oil which is in the same range as found by others [6, 29]. According to the high performance thin layer chromatography (HPTLC) analysis (Fig 2), a relatively high amount of free fatty acids are present in the unfractionated oil and this has been reported previously [9]. It can also be seen that both the neutral lipids (NL) and WE fraction were devoid of free fatty acids. The 49 mg difference in weight between the oil (322 mg) and isolated neutral lipids (273 mg) is therefore most likely due to free fatty acid fraction and a small amount of phospholipids in the oil. The 22 mg difference between the neutral lipids (273 mg) and isolated wax esters (251 mg) is most likely due to the absence of cholesterol, mono- and diacylglycerol in the isolated wax ester fraction.

The migration of the WE fraction on the HPTLC plate is as described by Henderson and Tocher [26] using this solvent system. In initial SPE experiments, we found that most of the NL were eluted with only 10 mL chloroform/isopropanol (2:1 v/v) and only a small fraction was obtained with an additional 10–20 mL (see Fig 1 in S1 File). However, to maximize the yield we eluted the NL with 30 mL. Similarly, we eluted the WE with 50 mL of heptane although the majority were eluted with the first 10 mL (see Fig 2 in S1 File).

### Saponification of wax esters and isolation of the fatty acids and fatty alcohols

The wax esters were hydrolyzed following Christie and Han [25] but using NaOH instead of KOH. Hydrolyses with KOH formed a soap like layer between the water and heptane layer that made it more difficult to recover the upper heptane layer containing the free fatty acids and free fatty alcohols. The method we used here showed to be effective in hydrolyzing the wax esters. In Fig 2, there are two clear separate bands visible, one corresponding to the free fatty acids and one band just below representing the free fatty alcohols. The free fatty acids band appears to consist of two bands with slightly different migration and this is probably due to different chain length and/or degrees of unsaturation of the different fatty acids present in the oil. No band is visible on the level of the wax esters, indicating that all wax esters were hydrolyzed.

After saponification, the free fatty acids and fatty alcohols present were separated from each other using SPE as shown in Fig 1 and described in the Materials and Methods section. A large fraction of the free fatty alcohols was eluted with the first 10 mL chloroform/isopropanol (2:1

**Table 1. Average yield of different lipid classes extracted 3 times from Calanus oil by means of solid phase extraction.** Values are given in mg and percentage weight of the oil with standard deviation.

| Lipid class | Average (mg) | STDEV | Average % | STDEV |
|---|---|---|---|---|
| Calanus oil | 322 | 19,1 | 100 | |
| Neutral lipids | 273 | 6,6 | 85 | 3,2 |
| Wax esters | 251 | 10,2 | 78 | 1,5 |
| Hydrolyzed wax esters | 153 | 10,1 | 48 | 3,6 |
| Free fatty alcohols | 75 | 7,4 | 24 | 3,5 |
| Free fatty acids | 63 | 6,5 | 20 | 3,2 |

v/v). To extract all fatty alcohols, however 30 mL was needed (see Fig 3 in S1 File). From on average 322 mg Calanus oil a yield of 75 mg free fatty alcohols was obtained (Table 1).

After conditioning the column with 20 mL heptane, the free fatty acids were eluted with 50 mL diethyl ether/acetic acid (98:2 v/v). Most fatty acids are eluted with 40 ml heptane in which non are present in the first 20 mL (see Fig 4 in S1 File). From on average 322 mg Calanus oil, 63 mg free fatty acids could be recovered (Table 1). The HPTLC analysis clearly shows that the FAOH and FFA present in the hydrolyzed wax esters have been separated (Fig 2).

## Composition of the fatty acids and fatty alcohols

Table 2 gives the fatty acid and fatty alcohol composition of the isolated wax esters and the isolated fatty acids and fatty alcohols after saponification. As expected, the results showed that almost

**Table 2. Relative composition of the fatty acids and fatty alcohols (%) in the isolated wax esters from Calanus oil and the isolated fatty acids and fatty alcohols fractions after saponification of the wax esters from the first isolation.**

| Fatty Acids | Wax ester | Isolated fatty acid | Isolated fatty alcohol |
|---|---|---|---|
| 14:0 | 9,8 | 19,7 | n.d. |
| 16:0 | 4,4 | 10,6 | n.d. |
| 18:0 | n.d. | 1,9 | n.d. |
| 16:1 n-7 | 2,6 | 5,2 | n.d. |
| 18:1 n-9 | 2,1 | 4,3 | n.d. |
| 20:1 n-7 | 0,9 | n.d. | 1,8 |
| 20:1 n-9 | 1,9 | 4,3 | n.d. |
| 20:1 n-11 | 0,4 | 0,8 | n.d. |
| 22:1 n-11 | 3,7 | 7,9 | n.d. |
| 18:2 n-6 | 0,6 | 1,2 | n.d. |
| 18:3 n-3 | 1,2 | 2,4 | 3,5 |
| 18:4 n-3 | 8,3 | 14,1 | n.d. |
| 20:3 n-6 | 1,5 | 0,8 | 2,6 |
| 20:4 n-3 | 0,8 | 1,3 | n.d. |
| 20:5 n-3 | 6,9 | 10,6 | n.d. |
| 22:5 n-3 | n.d. | 0,6 | n.d. |
| 22:6 n-3 | 3,5 | 4,5 | n.d. |
| Σ SFA | 14,2 | 32,2 | n.d. |
| Σ MUFA | 11,6 | 22,5 | 1,8 |
| Σ PUFA | 22,8 | 35,5 | 6,1 |
| Σ Fatty acids | 48,6 | 90,2 | 7,9 |
| **Fatty alcohols** | **Wax ester** | **Isolated fatty acid** | **Isolated fatty alcohol** |
| 14:0 | 0,6 | n.d. | 1,2 |
| 16:0 | 4,9 | n.d. | 9,1 |
| 16:1 n-7 | 1,2 | n.d. | 2,2 |
| 18:1 n-9 | 1,8 | n.d. | 1,6 |
| 20:1 n-9 | 12,6 | n.d. | 24,9 |
| 22:1 n-9 | 1,8 | n.d. | 2,4 |
| 22:1 n-11 | 22,5 | 0,8 | 46,0 |
| Σ Fatty alcohols | 45,4 | 0,8 | 87,4 |

Detection threshold is above 0,5%. n.d., not detected; SFA, saturated fatty acids; MUFA, monounsaturated fatty acids; PUFA, poly unsaturated fatty acids.

equal proportions fatty acids (48,6%) and fatty alcohols (45,4%) were found in the wax esters. Almost 80% of the identified fatty alcohols were monounsaturated, with eicosenol (20:1n-9) and docosenol (22:1n-11) being the dominating species. Of the polyunsaturated fatty acids, stearidonic acids (18:4 n-3), eicosapentaenoic acid (20:5 n-3) and docosahexaenoic acid (22:6 n-3) constituted about 30% of the fatty acids in the wax esters (Table 2). These results are similar to previous reported values [8, 9]. However, the composition of lipid classes and fatty acids in *C. finmarchicus* depends on the season, life cycle stage and geographic location [6, 30, 31]. It is therefore no surprise that the composition of the fatty acids and fatty alcohols in the wax esters found in this study is not exactly similar to the composition of the oil produced by the industry [8, 9]. The results from the fatty acid composition in the intact wax esters and the free fatty acid isolated from the hydrolyzed wax esters indicate that some oxidation may have occurred in the PUFAs, since the relative amount is lower in the isolated fatty acids than in the wax esters. Consequently, the relative amount of SFA has increased. Some oxidation may also have occurred in the MUFAs, but the relative amount is similar both in the intact wax esters and isolated fatty acids.

The established methods to analyze the oil in *C. finmarchicus* [11–13], focus on the determination of the composition of the lipids, rather than on extracting and separating the fatty alcohols and fatty acids for further use. Several methods have been described to extract policosanols from their natural sources, as reviewed by Shen *et al.* [32]. These methods include solvent extraction, transesterification and molecular distillation, supercritical carbon dioxide extraction, ultrasonic-assisted extraction, and saponification. As mentioned in the introduction, the previous described methods are excellent to study the fatty acid and fatty alcohol composition in calanoid species. But due to the production of FAMEs they are less suitable for the isolation of native, unesterified, fatty acid and fatty alcohol. It is possible to saponify the FAMEs to get free fatty acids. However, as can be seen in Table 1, the hydrolysis of the wax esters leads to a substantial weight loss from 251 mg to 153 mg. This leads us to suggest that an additional saponification step would further reduce the yield of the free fatty acids extracted from Calanus oil. To our knowledge, this is the first report of a simple, yet time consuming, method of semi-preparative isolation of the wax esters and the hydrolyzed components of wax esters, leaving the separated fatty acids and fatty alcohol unaltered. This allows them to be used for subsequent biological experiments.

## Supporting information

**S1 Table. Weight and percentage of Calanus oil weight of the extracted lipid classes from the 3 individual extraction rounds.**
(PDF)

**S1 File. Containing all supporting figures: Visualizations by means of TLC of the optimal elution volumes to extract the different lipid classes from Calanus oil.**
(PDF)

**S1 Raw images.**
(PDF)

## Acknowledgments

Zooca®, Tromsø, Norway provided the oil used in the study.

## Author Contributions

**Conceptualization:** Pauke Carlijn Schots, Guro Kristine Edvinsen, Ragnar Ludvig Olsen.

**Formal analysis:** Pauke Carlijn Schots.

**Investigation:** Pauke Carlijn Schots, Guro Kristine Edvinsen.

**Methodology:** Pauke Carlijn Schots, Guro Kristine Edvinsen, Ragnar Ludvig Olsen.

**Project administration:** Ragnar Ludvig Olsen.

**Supervision:** Guro Kristine Edvinsen, Ragnar Ludvig Olsen.

**Writing – original draft:** Pauke Carlijn Schots.

**Writing – review & editing:** Pauke Carlijn Schots, Ragnar Ludvig Olsen.

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
