## [Decision Letter · Decision Letter 0]

7 Feb 2023

PONE-D-22-35050A simple method to isolate fatty acids and fatty alcohols from wax esters in a wax-ester rich marine oilPLOS ONE

Dear Dr. Schots,

Thank you for submitting your manuscript to PLOS ONE. After careful consideration, we feel that it has merit but does not fully meet PLOS ONE’s publication criteria as it currently stands. As you may see below, there are significant concerns raised by the reviewers, in particular, on replication, and comparison with existing methods. Therefore, we invite you to submit a revised version of the manuscript that addresses the points raised during the review process. Please submit your revised manuscript by Mar 24 2023 11:59PM. If you will need more time than this to complete your revisions, please reply to this message or contact the journal office at plosone@plos.org. Please include the following items when submitting your revised manuscript:A rebuttal letter that responds to each point raised by the academic editor and reviewer(s). You should upload this letter as a separate file labeled 'Response to Reviewers'.A marked-up copy of your manuscript that highlights changes made to the original version. You should upload this as a separate file labeled 'Revised Manuscript with Track Changes'.An unmarked version of your revised paper without tracked changes. You should upload this as a separate file labeled 'Manuscript'.

We look forward to receiving your revised manuscript.

Kind regards,

Rishiram Ramanan

Academic Editor

PLOS ONE

Journal Requirements:

Reviewers' comments:

Reviewer's Responses to Questions

**Comments to the Author**

1. Is the manuscript technically sound, and do the data support the conclusions?

Reviewer #1: No

Reviewer #2: Yes

Reviewer #3: Partly

2. Has the statistical analysis been performed appropriately and rigorously? 

Reviewer #1: No

Reviewer #2: N/A

Reviewer #3: N/A

3. Have the authors made all data underlying the findings in their manuscript fully available?

Reviewer #1: Yes

Reviewer #2: Yes

Reviewer #3: Yes

4. Is the manuscript presented in an intelligible fashion and written in standard English?

Reviewer #1: Yes

Reviewer #2: Yes

Reviewer #3: Yes

5. Review Comments to the Author

Reviewer #1: The manuscript presents a simplified method to isolate fatty acids and fatty alcohols from a wax ester obtained from a zooplankton rich in monounsaturated fatty acids using HPTLC. The method seems to be really simple and not expensive. The procedures were described step by step and can be useful for other researchers. However, there are some parts that should be revised.

In general, there are several minor typos in the text.

Line 107: Include the volume of wax esters in chloroform that resulted in 240 mg

line 110 : it was described that the sample was kept at 90oC/90 min. Usually, an antioxidant uses to be added to avoid oxidative reaction at this step of the analysis.

Lines 143-154: include the reference of this part.

Line 157: It is important to detail which sub-type of standard was used to identify the fatty acids and fatty alcohols, since the information viable in the company web site has not been enough.

What is Akvaplan-niva ?

Table 1: It will be expected to have at least a triplicate of the assay. Without any replication it is not possible to obtain an error estimative. It is interesting to suggest what type of lipids correspond to the difference between waxe esters (239 mg) and neutral lipids (266 mg), and also between neutral lipids and calanus oil (300 mg).

It is important to better discuss the advantages and limitations of this method compared with others.

The manuscript is interesting but the lack of replication is not expected in an article about methodology alternatives. In addition, it is also expected a comparison with at least one classical method.

Thus, it is necessary to include replicate and compare with results obtained from other method.

Reviewer #2: The article describes a method for isolating free fatty acids and free fatty alcohols from wax esters of calanus oil. Although the paper is written very well and the descriptions are clear and sound, I doubt that the level of novelty justifies publication. The steps involved (extraction, saponification, SPE, checked by TLC) are well-described procedures for separating lipid classes and can be found in classic textbooks on lipid analysis (e.g. W.W.Christie…).

Furthermore, if this paper is aiming to optimize such a procedure, I would expect some data on method development. The authors merely present data of the (hopefully) optimized method. For method development, it would be essential to know, how the targets were defined (purity, solvent efficiency) and how the optimum was approached.

The authors claim that their method works without noticeable oxidation contrary to previous methods. How did the authors check this? I assume, that any oxidized fatty acid or alcohol would be retained in the SPE and thus noted as a reduced recovery.

As a minor comment, I wonder why the authors did not try to separate FFA from fatty alcohols as a soap before re-acidifying, instead of a SPE, given the fact that both molecular classes have similar polarities. At least this should be discussed.

Reviewer #3: The authors describe the preparation of mixtures of fatty acids and fatty alcohol from waxes present in Calanus finmarchicus oil. They propose the use of SPE columns to purify and isolate the waxes, followed by a basic hydrolysis with ethanol and NaOH. The main concerns relate to the methodology used and some of the results described (see specific comments).

Specific comments:

1.-Lines 64-65.- To study the health effects …it is necessary to separate the fatty acids and fatty alcohols …..

Why is it necessary? Provide a reference to support this claim. In any case, is the effect due to the functional class of each compound (alcohol or carboxylic acid) or to the presence of double bonds in the alkyl chain of both classes of compounds?

2.-Lines 66-67.- To our knowledge … no method … to separate the fatty acids and

fatty alcohols … without the formation of FAMEs.

What is the problem related to the preparation of FAMEs, the separation of the fatty alcohols from FAMEs and, finally, the recovery of fatty acids by hydrolyzing FAMEs?

3.- Lines 80-82.- … Phosphoric acid 99,99%, Potassium bicarbonate …..Toluene

Chemical names should be written in lowercase unless they are located after a period.

4.- Lines 107-108.- …wax esters …. dissolved in …. 1 M ethanolic NaOH to a concentration of 5 mg/mL…

There is an excess of ethanol related to the amount of water present (needed for the hydrolysis), do fatty ethyl esters not appear during this reaction?

5.- Various lines ..... minutes...hours ….

Time units should be indicated according IS rules: min, h…

6.- Lines 185-187.- free fatty acids band appear to consist of two bands is probably due to different degrees of unsaturation….

The different chain length of each acid can also lead to this situation. Please add this to the manuscript.

7.- Lines 223-224.- Table 2

How was the initial content of fatty acids and fatty alcohols in the crude wax determined?

How do the authors explain the differences between SFA, MUFA and PUFA for the fatty acids present in the wax and free fatty acids isolated after the SPE process and further saponification?

How do the authors determine the relative content from fatty acids and fatty alcohols as a whole? Did they consider the different response factors of each in the GC?

What is the fatty acid indicated (0.8 relative content) in the fatty alcohol section of the table?

8.- Figure 1

Did all SPE used contain the same chromatographic phase as described in line 88 (5 g Mega Bond Elute aminopropyl)?

6. PLOS authors have the option to publish the peer review history of their article (what does this mean?). If published, this will include your full peer review and any attached files.

Reviewer #1: **Yes: **Inar Castro Erger

Reviewer #2: No

Reviewer #3: No

---

## [Author Response · Author response to Decision Letter 0]

21 Mar 2023

We would like to thank all reviewers for their constructive feedback. We have gone through each comment and have changed the manuscript accordingly

Please see below the comments given by the reviewers. Directly below each comment is our response. The indicated line number refer to the manuscript with track changes. 

Reviewer #1: 

The manuscript presents a simplified method to isolate fatty acids and fatty alcohols from a wax ester obtained from a zooplankton rich in monounsaturated fatty acids using HPTLC. The method seems to be really simple and not expensive. The procedures were described step by step and can be useful for other researchers. However, there are some parts that should be revised.

In general, there are several minor typos in the text.

We have gone through the manuscript and have corrected several typos.

Line 107: Include the volume of wax esters in chloroform that resulted in 240 mg

We have included the average weight of wax esters obtained from the three extraction rounds in line 119 and mentioned that they are dissolved in 5mL chloroform. 

line 110: it was described that the sample was kept at 90oC/90 min. Usually, an antioxidant uses to be added to avoid oxidative reaction at this step of the analysis.

We did not add an antioxidant during the hydrolyzation step since Calanus oil contains naturally the antioxidant astaxanthin, which gives the oil its red color. 

Lines 143-154: include the reference of this part.

We have added on line 177-180 that the composition analyses of the fatty acids and fatty alcohols was done by the company Akvaplan-niva following their inhouse protocol that is based on methods described by J. Folch, M. Lees and Sloane Stanley; J. Biol. Chem., 226, 497 (1957) and Advances in Lipid Methodology - One (edited by W.W. Christie, Oily Press, Ayr), pp. 1-17 (1992).

Line 157: It is important to detail which sub-type of standard was used to identify the fatty acids and fatty alcohols, since the information viable in the company web site has not been enough.

The FAME standards are still available on the 2021-2024 catalogue on the company’s website. http://www.nu-chekprep.com/CATALOG%202021-2024.pdf. The fatty alcohol standard GLC 621 is no longer produced as a combination of alcohols, but each alcohol is available individually. The composition of GLC 621 is now given on line 173-177.

What is Akvaplan-niva ?

We have added on line 178-179 that Akvaplan-niva is a Norwegian company based in Tromsø, Norway.

Table 1: It will be expected to have at least a triplicate of the assay. Without any replication it is not possible to obtain an error estimative. 

We have now repeated the isolation two more times. The averages of the triplicate values are given in table 1, both in mg and percentage weight of the oil, with standard deviation. The individual values of each isolation are given in supplementary table 1. 

It is interesting to suggest what type of lipids correspond to the difference between waxe esters (239 mg) and neutral lipids (266 mg), and also between neutral lipids and calanus oil (300 mg).

We thank the reviewer for her comment. We had touched upon this topic in lines 188-191 but have extended this in lines 191-196. 

It is important to better discuss the advantages and limitations of this method compared with others.

We would again like to thank the reviewer for her comment. We have elaborated on the difference between the method we describe here and previously described methods (reference 10, 11, 12 and 13) and the advantages and limitations in the introduction in line 71-77 and in the results and discussion section in line 252-259. 

The manuscript is interesting but the lack of replication is not expected in an article about methodology alternatives. In addition, it is also expected a comparison with at least one classical method.

Thus, it is necessary to include replicate and compare with results obtained from other method.

With the 2 extra extractions performed, the average results are presented in table 1, and individual results in supplementary table 1, and the changes we have made to the manuscript we hope we have answered to the reviewer’s constructive feedback.

Reviewer #2: 

The article describes a method for isolating free fatty acids and free fatty alcohols from wax esters of calanus oil. Although the paper is written very well and the descriptions are clear and sound, I doubt that the level of novelty justifies publication. The steps involved (extraction, saponification, SPE, checked by TLC) are well-described procedures for separating lipid classes and can be found in classic textbooks on lipid analysis (e.g. W.W.Christie…).

We thank the reviewer for their comment. It is indeed true that the methods (SPE and TLC) used here are well-described procedures. We are, however, not aware of any publications describing a simple semi-preparative method for isolation of wax esters present in a wax-ester rich marine oil, and then separating the fatty alcohols and fatty acids constituting these wax esters. 

The novelty lies not in the method itself but in the application. As written in the introduction, on line 36-50, there is an increasing need for omega-3 fatty acids, and Calanus oil is a relatively new source of these lipids. In the last couple of years more and more studies on Calanus oil, and its potential health effects, have been done in both animals and humans (line 56-61). To study, in vitro, what part of the oil is responsible for what health effect it is necessary to separate the fatty acids from the fatty alcohols since the cells in our body will not be exposed to intact wax esters but rather to the separate fractions after hydrolyzes in the gut (line 62-69). To our knowledge we are the only lab today that is studying the individual effect of fatty acids and fatty alcohol. But with the growing demand for omega 3 fatty acids and the scientific interest in this oil by other labs worldwide, it can be expected that other labs will perform in vitro studies as well. It will then be of importance that the fatty acid and fatty alcohol fractions will be extracted in the same way to reduce potential confounding effects, through the method we describe here. We therefor do believe this paper justifies publication and we hope the reviewer agrees with us on that. 

Furthermore, if this paper is aiming to optimize such a procedure, I would expect some data on method development. The authors merely present data of the (hopefully) optimized method. For method development, it would be essential to know, how the targets were defined (purity, solvent efficiency) and how the optimum was approached.

We again thank the reviewer for their comment. Our main aim was to develop a method that would allow us to obtain the native, non-esterified fatty acids and fatty alcohols from wax esters in Calanus oil. As we show in figure 2, the lipid fractions we have obtained from the hydrolyzed wax esters appear pure. Regarding the optimization of our method, we have now added a supportive data file with TLC analyses from each extraction step described in figure 1, where we qualitatively show the presence of lipids eluted with different volumes of solvent to justify the volume choices in the method described here. 

The authors claim that their method works without noticeable oxidation contrary to previous methods. How did the authors check this? I assume, that any oxidized fatty acid or alcohol would be retained in the SPE and thus noted as a reduced recovery.

We do not claim that the method works without noticeable oxidation of fatty acids and fatty alcohols. We agree with the reviewer that if this occurs, oxidized fatty acids and fatty alcohols would be retained in the SPE and result in lower yields. We do, however, claim that we do not oxidize the fatty alcohol to their corresponding fatty acids during the isolation method, which allows us to use the fatty alcohols for further cell cultures. (line 29-33)

As a minor comment, I wonder why the authors did not try to separate FFA from fatty alcohols as a soap before re-acidifying, instead of a SPE, given the fact that both molecular classes have similar polarities. At least this should be discussed.

We have indeed tried to precipitate the FFA as a sodium salt and recover the soluble FaOHs, however we did not get a clear separation as we did with SPE. We have addressed this point now in line 135-138.

Reviewer #3: 

The authors describe the preparation of mixtures of fatty acids and fatty alcohol from waxes present in Calanus finmarchicus oil. They propose the use of SPE columns to purify and isolate the waxes, followed by a basic hydrolysis with ethanol and NaOH. The main concerns relate to the methodology used and some of the results described (see specific comments).

Specific comments:

1.-Lines 64-65.- To study the health effects …it is necessary to separate the fatty acids and fatty alcohols …..

Why is it necessary? Provide a reference to support this claim. In any case, is the effect due to the functional class of each compound (alcohol or carboxylic acid) or to the presence of double bonds in the alkyl chain of both classes of compounds?

As mentioned in line 62-69, it is necessary to separate the fatty acids from the fatty alcohols to study if, and how, these individual lipid classes, derived from wax esters, influence human health. In a previous publication, reference 24, (Schots PC, Pedersen AM, Eilertsen K-E, Olsen RL, Larsen TS. Possible Health Effects of a Wax Ester Rich Marine Oil. Frontiers in Pharmacology. 2020;11(961).) we have described the possible health effects of the different fatty acids and fatty alcohols in general that are present in Calanus oil. 

2.-Lines 66-67.- To our knowledge … no method … to separate the fatty acids and

fatty alcohols … without the formation of FAMEs.

What is the problem related to the preparation of FAMEs, the separation of the fatty alcohols from FAMEs and, finally, the recovery of fatty acids by hydrolyzing FAMEs?

We thank the reviewer for their comment. We have now explained on line 255-259 that the hydrolyzes of the wax esters led to a loss of material (as determined by weight). An additional hydrolyzes of the FAMES would further reduce the yield of the fatty acids. 

3.- Lines 80-82.- … Phosphoric acid 99,99%, Potassium bicarbonate …..Toluene

Chemical names should be written in lowercase unless they are located after a period.

We thank the reviewer and have changed the manuscript accordingly. 

4.- Lines 107-108.- …wax esters …. dissolved in …. 1 M ethanolic NaOH to a concentration of 5 mg/mL…

There is an excess of ethanol related to the amount of water present (needed for the hydrolysis), do fatty ethyl esters not appear during this reaction?

We believe they do not since this would have been visible on the TLC (figure 2)

5.- Various lines ..... minutes...hours ….

Time units should be indicated according IS rules: min, h…

We thank the reviewer and have changed the manuscript accordingly.

6.- Lines 185-187.- free fatty acids band appear to consist of two bands is probably due to different degrees of unsaturation….

The different chain length of each acid can also lead to this situation. Please add this to the manuscript.

We thank the reviewer and have changed the manuscript accordingly (line218).

7.- Lines 223-224.- Table 2

How was the initial content of fatty acids and fatty alcohols in the crude wax determined?

How do the authors explain the differences between SFA, MUFA and PUFA for the fatty acids present in the wax and free fatty acids isolated after the SPE process and further saponification?

How do the authors determine the relative content from fatty acids and fatty alcohols as a whole? Did they consider the different response factors of each in the GC?

What is the fatty acid indicated (0.8 relative content) in the fatty alcohol section of the table?

The fatty acid and fatty alcohol content in the wax esters and isolated lipid fractions was determined by the company Akvaplan-niva via an inhouse protocol as explained on line 161-180. 

Table 2 give values in percentage. The values of the different fatty acids in the isolated fatty acid fraction are about twice as high as those in the wax ester sample. This is because in the wax ester sample only half of the sample consists of fatty acids (48,6%), and the other half are fatty alcohols (45,4%). 

The fatty alcohol that is present with a relative content of 0,8 in the isolated fatty acid fraction is 22:1 n-11 (docosenol). Minor amounts of a few fatty acids are apparently detected in the fatty alcohol fraction, 1,8% 20:1 n-7 and 3,5% 18-3 n-3 and 2,6% 20:3 n-6. 

8.- Figure 1

Did all SPE used contain the same chromatographic phase as described in line 88 (5 g Mega Bond Elute aminopropyl)?

Yes, the same column (5 g Mega Bond Elute aminopropyl) was used for all different SPE steps in this procedure.

---

## [Decision Letter · Decision Letter 1]

13 Apr 2023

PONE-D-22-35050R1A simple method to isolate fatty acids and fatty alcohols from wax esters in a wax-ester rich marine oilPLOS ONE

Dear Dr. Schots,

Thank you for submitting your manuscript to PLOS ONE. After careful consideration, we feel that it has merit but does not fully meet PLOS ONE’s publication criteria as it currently stands. Therefore, we invite you to submit a revised version of the manuscript that addresses the points raised during the review process. The reviewers have largely agreed to accept the manuscript for publication except for a few minor edits. I invite the authors to submit the revised manuscript after addressing the reviewers' suggestions.

We look forward to receiving your revised manuscript.

Kind regards,

Rishiram Ramanan

Academic Editor

PLOS ONE

Journal Requirements:

Reviewers' comments:

Reviewer's Responses to Questions

**Comments to the Author**

1. If the authors have adequately addressed your comments raised in a previous round of review and you feel that this manuscript is now acceptable for publication, you may indicate that here to bypass the “Comments to the Author” section, enter your conflict of interest statement in the “Confidential to Editor” section, and submit your "Accept" recommendation.

Reviewer #1: All comments have been addressed

Reviewer #2: All comments have been addressed

Reviewer #3: All comments have been addressed

2. Is the manuscript technically sound, and do the data support the conclusions?

Reviewer #1: Yes

Reviewer #2: Yes

Reviewer #3: Partly

3. Has the statistical analysis been performed appropriately and rigorously? 

Reviewer #1: Yes

Reviewer #2: N/A

Reviewer #3: N/A

4. Have the authors made all data underlying the findings in their manuscript fully available?

Reviewer #1: Yes

Reviewer #2: Yes

Reviewer #3: Yes

5. Is the manuscript presented in an intelligible fashion and written in standard English?

Reviewer #1: Yes

Reviewer #2: Yes

Reviewer #3: Yes

6. Review Comments to the Author

Reviewer #1: I only suggest two minor revisions.

Line 60: making

Line 122: It is important to include the authors answer in the text. For example: “No antioxidants were added during the wax esters hydrolysis because Calanus oil contains naturally the antioxidant astaxanthin”.

Reviewer #2: In my view, the authors have clarified all points satisfactorily. There is just one point that popped in to my eyes: In the description of figures, it should read."...from bottom to top.." - not top to bottom (all figures and supportive figures)

Reviewer #3: I consider that the authors have only partially addressed my previous comments. I attach a document with my considerations.

7. PLOS authors have the option to publish the peer review history of their article (what does this mean?). If published, this will include your full peer review and any attached files.

Reviewer #1: **Yes: **Inar Alves Castro

Reviewer #2: No

Reviewer #3: No

---

## [Author Response · Author response to Decision Letter 1]

21 Apr 2023

We would like to thank all reviewers again for their constructive feedback and have changed the manuscript accordingly.

Please see below in red our response to each point raised by the reviewers. The indicated line number refer to the manuscript with track changes. In addition, some minor linguistic changes have been made on line 44, 69, 73, 109, 171, 173 and 179.

Reviewer #1:

Line 60: making: 

We have changed the sentence on line 60, making it one sentence instead of two.

Line 122: It is important to include the authors answer in the text. For example: “No antioxidants were added during the wax esters hydrolysis because Calanus oil contains naturally the antioxidant astaxanthin”.

We have added the sentence as suggested by the reviewer on line 122-123

Reviewer #2:

Reviewer #2: In my view, the authors have clarified all points satisfactorily. There is just one point that popped in to my eyes: In the description of figures, it should read."...from bottom to top.." - not top to bottom (all figures and supportive figures).

This is indeed correct, and we thank the reviewer for pointing this out. We have changed the figure legend as the reviewer mentions in the figure text of the raw images. In the figure text in the manuscript and the supplementary data file the different lipid classes are identified with a letter and we therefore did not add the phrase “from bottom to top”. 

Reviewer #3

Line 138 …..n in our set-up using NaOH as a salt.

NaOH cannot be defined as a salt from a chemical point of view. Of course, it can modify the ionic strength of a solution, but as a base. Please amend this sentence.

This is indeed correct, and we have changed the sentence on line 136-137 accordingly.

2.- Table 2

The authors do not explain the differences between SFA, MUFA and PUFA for the fatty acids present in the wax and the free fatty acids isolated after the SPE process and further saponification. According to the percentages indicated in Table 2, the percentages referring to 90.2 % of the total FFA isolated should be 26.35, 21.5 and 42.31 for SFA, MUFA and PUFA, respectively considering the initial percentages of fatty acids found in the waxes. However, the percentages found are 32.2, 22.5 and 35.5, which show an increase in SFA and a decrease in MUFA and, even more, in PUFA. In my opinion, this clearly indicates a higher loss of unsaturated versus saturated fatty acid during the sample processing. The authors should clearly explain this in the manuscript.

Also this is correct and we thank the reviewer for pointing this out. We have added a part on line 243-248 to explain this.

---

## [Editor Report · Decision Letter 2]

2 May 2023

A simple method to isolate fatty acids and fatty alcohols from wax esters in a wax-ester rich marine oil

PONE-D-22-35050R2

Dear Dr. Schots,

We’re pleased to inform you that your manuscript has been judged scientifically suitable for publication and will be formally accepted for publication once it meets all outstanding technical requirements.

Kind regards,

Rishiram Ramanan

Academic Editor

PLOS ONE
---

## [Editor Report · Acceptance letter]

5 May 2023

PONE-D-22-35050R2 

A simple method to isolate fatty acids and fatty alcohols from wax esters in a wax-ester rich marine oil 

Dear Dr. Schots:

I'm pleased to inform you that your manuscript has been deemed suitable for publication in PLOS ONE. Congratulations! Your manuscript is now with our production department. 

Kind regards, 

on behalf of

Dr. Rishiram Ramanan 

Academic Editor

PLOS ONE